# A Literature Review of Modelling and Experimental Studies of Water Treatment by Adsorption Processes on Nanomaterials

**DOI:** 10.3390/membranes12040360

**Published:** 2022-03-24

**Authors:** Qusai Ibrahim, Leo Creedon, Salem Gharbia

**Affiliations:** Institute of Technology Sligo, Ash Lane, F91 YW50 Sligo, Ireland; qusai.ibrahim@mail.itsligo.ie (Q.I.); creedon.leo@itsligo.ie (L.C.)

**Keywords:** density functional theory, adsorption, nanomaterials, wastewater treatment, simulation

## Abstract

A significant growth in the future demand for water resources is expected. Hence researchers have focused on finding new technologies to develop water filtration systems by using experimental and simulation methods. These developments were mainly on membrane-based separation technology, and photocatalytic degradation of organic pollutants which play an important role in wastewater treatment by means of adsorption technology. In this work, we provide valuable critical review of the latest experimental and simulation methods on wastewater treatment by adsorption on nanomaterials for the removal of pollutants. First, we review the wastewater treatment processes that were carried out using membranes and nanoparticles. These processes are highlighted and discussed in detail according to the rate of pollutant expulsion, the adsorption capacity, and the effect of adsorption on nanoscale surfaces. Then we review the role of the adsorption process in the photocatalytic degradation of pollutants in wastewater. We summarise the comparison based on decomposition ratios and degradation efficiency of pollutants. Therefore, the present article gives an evidence-based review of the rapid development of experimental and theoretical studies on wastewater treatment by adsorption processes. Lastly, the future direction of adsorption methods on water filtration processes is indicated.

## 1. Introduction

Water is arguably the main life source on planet earth and is vital for preservation of the modern world. In many parts of the world, water exploitation in agriculture and land development has led to significant economic progress [1]. On the other hand, water pollution is the biggest challenge facing the world and causing the destruction of water resources around the world [2]. This is because of its pollution with oil, chemicals, industrialization, human consumption, leakage of sewage lines, or the absence of these lines [3]. In addition, according to the United Nations (UN) report, some countries in the Middle East region suffer from water scarcity [1]. Moreover, human activities such as overgrazing, cultivation in wet areas, and discharging waste into rivers has changed the water flow patterns in rivers in terms of quantity, quality and flow times [4]. Changing flow quantities can affect water quality significantly in ground and surface water such as rivers [5]. An example of this is in Rajasthan, India due to a very large increase in groundwater pumping [6]. This led to a decrease in water quality due to an increase in the concentration of fluoride above normal levels causing disease in bones and teeth [6]. In other cases, when the consumption of surface water sources is very high, such as rivers and lakes, the decrease in water levels leads to an increase in the concentration of minerals [7]. These problems have led to a lack of safe drinking water for more than a third of the world population [8]. In addition, it creates the need to look for high quality water by converting low quality water to pure water. Some of the latest technologies that revolutionised the field of water filtration in recent years are membrane-based methods and photocatalysis degradation of organic pollutants. In this review, we mainly focused on the role of the adsorption process in both membrane technology and photocatalytic degradation of organic pollutants by experimental and simulation methods.

### 1.1. Water Filtration

Filtration is the process of removing solid particles or gases from liquids through filter media which allow the liquid to pass through and blocking solid and gases particles [9]. In filtration processes, filtered water goes one way and other particles collected through the filter area go in a different route [10]. Filtration processes have received wide attention since the 18th century to purify polluted water to obtain freshwater sources for drinking and commercial uses [11]. Unfortunately, many systems which have been developed to meet the needs of water purification were unsuccessful due to drawbacks in each method. In the 18th century, filtration was adopted to remove undesirable particles from water such as copper and lead [12]. Sponge filter was the first water filtration device discovered around the mid-1700s, and Joseph Amy obtained a patent for that device [13]. Furthermore, water purification devices were provided for the first time for domestic use in 1750 [14]. In addition, the first known slow sand filtration system was made in 1804 by John Gibb in Scotland and was very effective in the removal of bacteria, viruses, and heavy metals to produce drinking water [15]. There were great developments in water filtration systems in the 19th and 20th centuries such as the development of the reverse osmosis (RO) method [16], membrane technology [17], and photocatalytic degradation of organic pollutants [18]. The RO method forces water to move through a semi-permeable membrane by an external pressure [19,20]. A larger volume of water subsequently passes through the membrane compared to the volume of dissolved salts or organic molecules [19]. It is capable of removing 99% of ions, particles, collides, and bacteria from the feed water [19].

Energy consumption is a key factor which affects the freshwater production cost in the RO process [21]. One of the main factors for the high energy consumption in the RO method is the high-pressure pump which consumes almost 74% of total energy consumption in RO systems [22]. It has become more efficient for water desalination with the advance of nanotechnology [23]. The level of RO energy consumption has declined in the past 40 years due to membrane technological improvements offering higher permeability [24]. Membrane technology is a general term for several different separation processes that depend on a semi-permeable membrane for gas separation and the removal of undesirable ions or molecules from liquids [25]. The semi-permeable membranes with specific pore size are fixed between two media to block particles at the molecular or ionic level by a driving force based on the pressure difference between the two sides [26,27].

Nanocomposite membranes have attracted researchers to develop better materials to achieve remarkable properties such as selectivity, mechanical properties, and enhancement of membrane performance in water desalination [28]. For instance, mechanical properties (including Young’s modulus, bulk modulus, and shear modulus) and selectivity of graphene membrane have improved after the combination with titanium dioxide (TiO_2_) as indicated in a previous study [29,30,31]. Nanofiltration is a process depending on an external pressure force in which molecules and particles in a range between 1–2 nm are rejected by the membrane which make it one of the most widely used membrane processes in desalination and wastewater treatment [32]. However, researchers have focused on developing a variety of nanoporous membranes which have open pores diameter ranging between 1 to 100 nm for water desalination such as graphene and molybdenum disulphide (MoS_2_) [33]. These two nanoporous membranes achieved a high water permeability and salt rejection performance [34,35]. Graphene membrane has contributed to other applications in the water filtration field. In 2016, Cohen-Tanugi and colleagues synthesized a multilayer graphene membrane and it proved to be more economical than a single layer graphene membrane [36]. In addition, the multilayer graphene membrane can offer higher salt rejection than a single membrane with high permeability [36]. However, these promising results attracted researchers to synthesize other nanocomposite materials such as multilayer MoS_2_ membrane, and multilayer graphene oxide (GO) membrane [37]. The multilayer membranes are expected to offer more flexibility in terms of membrane productivity, membrane lifetime, and their performance in filtration processes [38]. In addition, a lot of studies mentioned the contribution of spinel ferrites (MFe_2_O_4_) with metal oxides, MFe_2_O_4_/carbon-based materials, MFe_2_O_4_/polymers, and MFe_2_O_4_/metal nanoparticles for the photocatalytic degradation of dyes and other inorganic pollutants as mentioned in previous studies [39,40,41]. In addition, Zeolite nanostructured membranes have been widely used for the removal of hazardous chemicals from a contaminated water solution as shown in previous studies [42,43,44].

On the other hand, nanocomposite materials such as MoS_2_ showed distinctive properties in the photocatalytic degradation of organic pollutants [45]. This is because of its unique electronic [46] and optical properties [47], and its size-dependent bandgap [48]. In 2020, Yan et al. synthesized MoS_2_/TiO_2_ nanotube composite for efficient water disinfection using anodic oxidation method and hydrothermal method [49]. The results showed an excellent photocatalytic disinfection under visible light irradiation for the removal of bacteria up to 98.5% with the possibility of reusing the nanocomposite and recycling it efficiently [49]. Photocatalysis is an eco-friendly technique for the removal of many different pollutants such as nitrogen oxides (NO_x_), pesticides and other organic pollutants using ultraviolet (UV) light or sunlight [50,51]. TiO_2_ is considered to be the most promising material for the photocatalytic removal of organic pollutants [52]. This is because of the high photocatalytic activity of TiO_2_ which make it environmentally friendly [53]. Furthermore, the number of pores located in the TiO_2_ surface that are generated by photons make it easy to be harvested by free electrons from outside the surface of TiO_2_ [53]. In addition to wastewater treatment, photocatalysis has many other applications such as storing energy [54], air purification [55], and antifouling by preventing the nonspecific interaction between membrane surface and foulants [56]. However, photocatalytic reactions proved their effectiveness in degradation of organic pollutants in non-toxic filtration medias, without using chemicals [57]. One of the important steps in the photocatalytic process is the adsorption between the reactant substances with the catalyst surface [58]. The adsorption process affects the efficiency of the photocatalysis process because it mainly depends on the catalyst absorption of light source [59,60]. Therefore, it is one of the most important water filtration processes under development that controls filtration [61]. Moreover, there are many advantages of adsorption process such as being inexpensive [62], fast [63], and simple in operation and implementation [64]. It has many applications in wastewater treatment to remove pollutants such as bacteria and heavy metals and has received considerable attention from researchers [65].

### 1.2. Adsorption Techniques

In water filtration, adsorption is the process of removing organic pollutants from wastewater in which a binding energy is present between the molecules of the substance (ions or atoms) with other surfaces by chemical or physical attraction [66]. It is an important process in catalysis [67], chemical engineering [68], and material science [69]. There are two main types of adsorption process: physical adsorption (physisorption) and chemical adsorption (chemisorption) [70]. Physical adsorption occurs by adsorbing gas molecules onto a solid surface using low intensity forces called van der Waals force [71,72]. London-van der Waals is a dispersion force acting between microscopic non-deformable bodies such as atoms and molecules [73]. There are many applications of physisorption process such as: hydrogen storage [74], acoustic wave sensors [74], gas sorption [75] and water filtration [76]. Furthermore, physical adsorption improves membrane efficiency in terms of water flux, hydrophilicity, and antifouling [77]. For instance, Peng and colleagues modified the performance of PVDF (polyvinylidene fluoride) microfiltration membrane for water filtration by a strong physisorption of amphiphilic copolymers experimentally [78]. The results showed a high improvement in the permeability of PVDF membrane with better antifouling properties without any effect on the membrane structure [78].

On the other hand, in chemical adsorption the bonding occurs between the surface molecules of a metal with high energy and another substance (adsorbate) in contact with it, which may be a liquid or a gas [79]. The bonds formed are comparable in strength to ordinary chemical bonds and are much stronger than the van der Waals forces characteristic of physical adsorption [79]. Chemisorption has been widely used in industrial wastewater treatment for the removal of heavy metals [80,81,82]. For instance, Liu et al. investigated the chemical adsorption behaviour in the removal of arsenic by experimental and simulation techniques using microporous metal-organic framework (MIL-125(Ti)). The results showed a fast and efficient removal of arsenic with low concentrations by chemical adsorption [83]. Most studies have not focused on adsorption kinetics by either ignore it, excluded it, or by assuming the adsorption interaction approaches the equilibrium [84,85,86]. However, Luo et al. have studied the photocatalytic degradation kinetics of graphitic carbon nitrate (g-C_3_N_4_) for contaminant removal by using a 36 W LED light (λ = 400 nm) in a dark chamber [87].

There are many factors that affect the performance of chemical and physical adsorption processes in water filtration such as: atmospheric and experimental conditions [88], contact time between the adsorbate and adsorbent [89], and particle size [90]. It was noted that atmospheric conditions significantly affect the effectiveness of physical adsorption process in terms of absorption capacity [88]. The adsorption process generally is an exothermic process where the energy is expelled in the form of heat or light [91]. Therefore, physical adsorption is more efficient at lower temperatures because when the temperature rises, the ability of the material to absorb reduces as indicated in previous studies [92,93,94]. On the other hand, chemical adsorption range increases with increasing temperature to a certain extent and then begins to decrease [95]. In addition, chemical and physical adsorption increases with increasing gas pressure to a certain extent until saturation is reached [96]. Contact time and particle size have a major effect on the efficiency of both types of adsorption process [89]. Zhang et al. noted that the adsorption of methylene blue (MB) from aqueous solution increased with contact time with the adsorbent [89]. Along the same lines, Laabd et al. studied the adsorption capacity of polyaniline (PANi) film for the removal of trimellitic and pyromellitic acids by experimental and density functional theory (DFT) methods [97]. They estimated the optimum experimental conditions for the adsorption of the acids in terms of contact time, pH, initial concentration, and temperature. Meanwhile, they investigated the results by using first principle DFT calculations and studied the physical interactions between the adsorbate and adsorbent surface molecules [97]. In addition, Bergaoui et al. studied the adsorption mechanism of methylene blue (MB) onto organo-bentonite [98]. The results showed a high MB removal with a maximum adsorption capacity up to equal to 321 mg/g. However, adsorption technique enhanced the performance of photocatalysts based membranes in wastewater treatment as reported in previous studies [99,100,101]. For instance, Zhang et al. modified Ag@BiOBr/AC/GO membrane system for efficient removal of rhodamine B (RhB) by membrane separation and high adsorption capacity [99].

So far, molecular dynamic (MD) and DFT calculations have been widely used by researchers to investigate and predict experimental results and save money, as well as acquiring faster results [102]. Adsorption process and its contribution in water and wastewater treatment has been studied experimentally and investigated by simulation in 70 publications which will be reviewed in this paper. Figure 1 shows the rapid increase in recent years in the number of publications related to water treatment by adsorption process using experimental and simulation methods.

### 1.3. Computational Methods

Computer simulation methods are compatible with experimental work in a laboratory to serve as a bridge between laboratory experiments and theoretical calculations [103]. Furthermore, simulation is an approximate emulation for an integrated complex system to analyse the behaviour and performance of the system over time [104]. There are many applications for simulation such as: manufacturing [105], economics [106], safety engineering [106], and simulation of technology for performance or optimization [107]. Simulation has many advantages such as validating the results obtained by other analytical methods, and sometimes finding unexpected phenomenon while studying the behaviour of the system [108]. MD and DFT simulation methods have been used in many studies in fabrications and modifications of membrane structure for water desalination, gas separation, electrolysis, and many other applications [35,109,110]. In addition, they have been used to investigate the experimental results of the adsorption process as indicated in previous studies [111,112,113].

#### Calibration and Validation

Due to the significant increase in the use of computer simulation methods in the 21st century, it was necessary to find ways to verify the data issued by simulation programs. Therefore, many models have been modified to measure the error ratio between the data obtained experimentally and data obtained by simulation such as SWAT (Soil and Water Assessment Tool) [114]. SWAT is used to calibrate and validate the data by a procedure using the shuffled complex evolution method [114]. Calibration “is a major element to this evaluation and refers to the estimation and adjustment of model parameters to improve the agreement between model output and a data set” [115]. However, validation “is a model using parameters that were determined during the calibration process and comparing the predictions to observed data not used in the calibration” [115]. Recently, researchers’ dependence on calibration and validation of their results have been gradually increased. Therefore, the number of publications that studied adsorption process in wastewater treatment using combined experimental and simulation methods increased at the beginning of 2017. Figure 2 shows the number of publications in water treatment by adsorption using experimental and simulation methods in the last 10 years.

For instance, de Oliveira and colleagues studied the adsorption process of 17β-estradiol in graphene oxide through methanol co-solvent experimentally and through simulation using SIESTA code (Spanish Initiative for Electronic Simulations with Thousands of Atoms) [113]. The DFT model was calibrated using the experimental findings and was in good agreement after studying the electronic density of state (DOS) and the interactions between graphene oxide with methanol molecules [113]. Along the same lines, Zhu et al. studied the degradation of bisphenol using highly efficient heterogeneous Fenton catalysts (CNTs/Fh) by experiment and simulation [116]. The main aim of this study was to accelerate Fe(III)/Fe(II) cycling by carbon nanotubes (CNTs) and they had an excellent agreement in the experimental and DFT results [116]. More detail about the fabrication and characterization of the nanocomposite materials experimentally and by simulation are given in Section 2.

## 2. Synthesis and Simulation of Nanomaterials

In the last decade, synthesis and simulation of nanomembranes have received vast attention and have been widely studied by researchers [117]. Figure 3 shows the most common methods used in the synthesis of nanomaterials.

### 2.1. Nanomembranes

Synthesis and characterization of nanomembranes have received wide attention since the 18th century [118]. During the 18^th^ century, membranes were under fabrication, functionalization, and modification at the laboratories without any commercial use [119]. Since 2004, membrane experimental designs have increased and the number of materials available for these experiments has increased [118]. One of the famous designs was by Jani and colleagues which designed nanoporous anodic aluminium oxide membranes with desired functions, parameters and properties [120]. Similarly, Mei and colleagues fabricated ultrathin AlN/GaN porous crystalline nanomembranes with different layouts including tubes, spirals, and curved sheets [121]. The structural, morphological and chemical properties of nanomembranes will be characterizing using analyses such as X-ray diffraction (XRD), scanning electron microscope (SEM), thermal electron microscope (TEM) and X-ray photoelectron spectroscopy (XPS), Raman spectroscopy, etc. [122]. The optical and electrical properties of the nanomembranes also will be analysed using UV-Vis diffuse reflectance spectroscopy (UV-Vis DRS) [123]. “Nanomembranes are synthetic structures with a thickness less than 100 nm and the aspect of surface-area-volume ratio increases to at least a few orders of magnitude” [124]. Nanomembranes can be classified based on surface chemistry, bulk structure, morphology, and production method [119]. Nanomembranes have been widely used in many applications such as water and wastewater treatment [125], biomedical applications [126], and smart energy storage devices [127]. In this section, we demonstrate the most common methods used in the synthesis and simulation of nanomembranes.

#### 2.1.1. Synthesis of Nanomembranes

Synthesis is a term for producing nanostructured materials including organic, inorganic, and hybrid nanomembranes [128]. It exploits the special physicochemical properties of ionic fluids to control transit and growth [129]. Many methods have been used for the synthesis of nanomembranes such as modified Hummers’ method [130], solvothermal method [131], and solvothermal chemical deposition [132]. Modified Hummers’ method is one of the most common methods used for the synthesis of nanomembranes such as graphene oxide (GO) [133]. It was developed in 1958 with many advantages such as being safer, faster, and a more efficient method for producing graphite oxide [134]. The chemical method can generate graphite oxide through the addition of potassium permanganate to a solution of graphite, sodium nitrate, and sulfuric acid [129]. However, for the synthesis of other nanomembranes such as molybdenum disulphide, the microwave-assisted route has been used [135]. The microwave-assisted route is “a unique and simple technique for fast and efficient processing of materials with higher reproducibility” [136]. It has drawn attention due to its homogeneous heating, fast kinetics, high phase purity, and high yield rate of products in relatively short time [136]. Table 1 shows nanomembranes synthesized by different synthesis methods.

As shown in Table 1, 11 types of nanomembranes have been synthesized by using different materials. Graphene and graphene oxide (GO) were the most synthesized nanomembranes by using Hummers’ method because of their widespread use in water and wastewater treatment. However, other nanocomposites such as nitrogen doped carbon (CNs), are synthesized by using chlorination of Ti(C_0.7_N_0.3_) at various temperatures resulting in well-developed micro-pores and small meso-pores with uniform pore structures.

#### 2.1.2. Simulation of Nanomembranes

Density functional theory (DFT) is a computational simulation method used in chemistry, physics, and materials science for the calculation of the mechanical and electronic properties of atoms and molecules [155]. There are many simulation software used for DFT calculations such as Material Studio, Vienna Ab initio Simulation Package (VASP), and GAMESS, etc. The simulation software have been used by researchers and engineers to improve the performance of materials in many applications including pharmaceuticals, catalysts, polymers and composites, metals and alloys, batteries and fuel cells [156]. They have many advantages such as developing new cost-effective materials with better performance and more efficiently than with test and experimentation alone [157]. Material studio is a three-dimensional (3D) modelling and simulation software developed and distributed by BIOVIA to allow researchers in material science and chemistry to understand the behaviour and relationships of a material’s atomic and molecular structure [156]. Similarly, VASP, Gaussian 09, and GAMESS have been used for atomic scale materials modelling using DFT with different functional groups including (B3LYP) and different methods such as the projector augmented wave method (PAW), and Perdew-Burke-Ernzerhof (PBE) method [158]. PAW and PBE methods are both efficient for the electronic structure calculations of large systems [159]. Furthermore, they are used to improve the accuracy of the electrical and electronic calculations for magnetic materials, alkali and alkali earth elements [160].

Figure 4 shows simulation software used to produce nanomembranes with the number of publications using each software.

As shown in Figure 4, Gaussian 09 and VASP contributed to the simulation of 15 nanomembranes out of 22 nanomembranes in this review paper. 8 of the 15 nanomembranes were graphene or graphene oxide (GO). This is due to the high accuracy in the simulation of graphene and GO nanomembranes by these simulation software as indicated in previous studies [161,162]. In addition, molybdenum disulphide (MoS_2_) nanosheet has been simulated by using VASP with PAW simulation method, while other nanocomposites such as Zn–Fe LDH, and (CF/BiOBr/Ag_3_PO_4_) cloth, have been simulated by using Material Studio with DMol3 and GGA-PBE codes, respectively. Table 2 shows the simulation software and methods used for simulation of nanomembranes.

As shown in Table 2, PAW, PBE, and B3LYP are the most common methods used for the DFT calculations of nanomembranes. These calculations are performed based on the solution of Kohn-Sham equations by PAW method. On the other hand, the exchange-correlation functional model, and the Thomas, Yoon–Nelson, and Adams–Bohart model have been solved by B3LYP and PBE method, respectively. Along the same lines, these methods (PAW, PBE, and B3LYP) have been used for the simulation of nanocomposite materials as explained later in Section 2.2.2.

### 2.2. Nanocomposites

Nanocomposite materials “are composed of several multiple nanomaterials entrapped within a bulk material, which may comprise a combination of a soft and a hard nanomaterial, two soft nanomaterials, or two hard nanomaterials” [169]. They are characterized by their very small size, measured in nanometres [170]. Nanocomposite materials have attractive properties resulting from the combination of inorganic or organic components at the molecular level [171,172]. There are many applications of nanocomposite materials in wastewater treatment [173], energy storage [174], drug delivery [175], and for biomedical purposes [176]. In wastewater treatment, nanocomposite materials have been widely used to treat surface water, sewage, and ground water [177]. By 2009, nano-processing technologies were documented at 44 cleaning sites around the world, most of them in the United States [178]. The synthesis of these nanocomposites received wide attention by the researchers in the last decade [179]. In this section, we demonstrate the most common methods used in the synthesis of nanocomposite materials.

#### 2.2.1. Synthesis of Nanocomposites

For synthesis and characterization of these nanomaterials, many methods have been used including the hydrothermal method [180], chemical vapor deposition (CVD) [181], and one-pot synthesis [182]. The hydrothermal method is one of the most common methods used in the synthesis of nanocomposites [180]. Figure 5 shows the percentages of the number of publications reviewed in this review paper by each experimental method. As shows in Figure 5, the hydrothermal method has been used in more than 56% of the publications reviewed.

##### The Hydrothermal Method

Hydrothermal synthesis is a method that uses very high temperatures ranging from room temperature to much higher temperatures to synthesize nanomaterials [183]. It was given the name “hydrothermal” because water is used as the solvent [184]. The hydrothermal method was first discovered in the 19th century [185]. It has been widely used by researchers and the first publication on this method appeared in 1813 [185]. The publication was about “Synthesis and Characterization of Zinc Tin Sulphide (ZTS) Thin Films via Chemical Bath Deposition Route” [185]. Hydrothermal synthesis has many advantages over other synthesis methods including “top down” method, “bottom up” method, and sol-gel method such as being an environmentally friendly, low-cost synthesis method, its simplicity, and the production of high-quality one-dimensional (1D) nanostructures [186,187,188,189]. However, there are some disadvantages for this method: taking a long time in the production process, corrosion, and difficulty in recycling and regenerating the catalysts [184,190]. Recently, hydrothermal synthesis has been used in several applications in science such as food and nutrition, organic chemistry, environmental safety, and energy applications [191,192]. For instance, Zhu et al. synthesized a highly efficient heterogeneous Fenton catalyst (CNTs/Fh) for the degradation of (bisphenol A) by using a hydrothermal method [116]. Similarly, Wang et al. synthesized a pyridinic-N doped graphene/BiVO_4_ nanocomposite (N-rGO/BiVO_4_) by hydrothermal method with a great potential for the removal of pollutants from wastewater [193]. Table 3 shows the nanocomposite materials synthesized by the hydrothermal method in the last decade.

As shown in Table 3, the hydrothermal method has been used in the synthesis of different nanocomposite materials including titanium dioxide (TiO_2_) nanoflowers, nanomaterials with carbon nanotubes (CNTs), and metal oxides with carbon. The reason for the wide use of the hydrothermal method is its advantages over others in the ability to create crystalline phases, even those which are not stable at the melting point [218]. For instance, Zhao and colleagues synthesized TiO_2_ nanoflowers (TNFs) using hydrothermal and calcination treatments [199]. The results showed a strong photocatalytic capability, and satisfactory recycled stability of the TNFs, which enhances their value for practical applications in water purification [199]. Along the same lines, Cheng et al. synthesized a titanate nanotube supported TiO_2_ (TiO_2_/TiNTs) using the hydrothermal method [200]. The results showed that TiO_2_/TiNTs significantly eliminated the toxicity of phenanthrene and can greatly decrease the potential risks of phenanthrene to aquatic organisms [200].

##### Chemical Vapor Deposition

Chemical vapor deposition (CVD) is a coating process that is defined as a method to produce solids with high purity by using thermally induced chemical reactions at the surface of a heated substrate [219]. CVD has many applications in medicine [220], electronic applications [221], and chemical industries [222]. It has many advantages over other synthesis methods such as the ability to deposit a wide variety of materials with very high purity [223]. The CVD method started in the 19th century with the production of lamp filaments. Then, Van Arkel in the 20th century deposited metals from the gas phase for application in the lamp industry [224,225]. The CVD method has three different types based on the conditions of the process classified by applied pressure [226], physical properties of the vapor [227], and plasma methods [228]. It has been used in the production of several materials including monocrystalline, polycrystalline, amorphous, preparation of carbon nanotubes (CNTs) and carbon nanofibers [221,229]. In addition, CVD is famous for producing semiconductors such as the synthesis of 2D Tungsten disulphide (WS_2_) monolayer [230]. Table 4 shows nanocomposite materials synthesized by the chemical vapor deposition (CVD) method.

In this review paper, the CVD method contributed in 7% of the total number of publications as shown in Figure 5. Mainly, it has been used for the synthesis of carbon nanotubes (CNTs). As shown in Table 4, the CVD method has been used for the combination of Co_3_O_4_, and COOH with CNTs. At a sufficiently high temperature, carbon source (hydrocarbon gas) decomposes with the catalyst in a tubular reactor [233]. By using CVD, Yang et al. synthesized a vertically aligned carbon nanotube hybrid membrane for gas separation [164]. The results showed an excellent separation membrane with high conductivity and resistance stability after 50 cycles of tensile deformation [164]. In addition, Zhang et al. studied the adsorption of lead (Pb^2+^) on oxidized (O-CNTs) and graphitized multi-walled carbon nanotubes (G-CNTs) synthesized by the CVD method [163]. The results showed a high stability in the adsorption mechanism of Pb^2+^ [163].

##### One-Pot Synthesis

One-pot synthesis is a process that is used to improve the efficiency of chemical reactions and focuses on the reduction of number of steps of chemical reactions in one single reaction flask [234]. It is a hydrothermal approach based on a general phase transfer and separation mechanism which occurs at interface of water, solution, and solid phases [190]. Das et al. reported in their book that the one-pot synthesis method has many advantages such as “saving time and resources, improves the efficiency of a chemical reaction, and offers better chemical yield” [190]. An example of one-pot synthesis is the synthesis of highly stable CsPbBr_3_@SiO_2_ Core–Shell Nanoparticles [182]. The reported method showed that the formation rates, determined by reaction temperature, precursor species, pH value, etc., of both CsPbBr_3_ and SiO_2_ are critical for the successful preparation of core–shell NPs [182]. Table 5 shows nanocomposite materials synthesized by the one-pot synthesis method.

As shown in Table 5, Ren and his colleagues synthesized a 3D porous sulphur and nitrogen co-doped graphene aerogel (SN-rGO-A) for the degradation of Rhodamine B (RhB) [165]. The results showed that sulphur and nitrogen co-doping could synergistically enhance the catalytic performance for activating peroxydisulfate (PDS) compared to the original and N doped graphene aerogels [165]. Along the same lines, Li and colleagues synthesized a series of catalysts by using ZIF-67 for the removal of bisphenol A (BPA) and total organic carbon (TOC) [235]. The results showed an excellent degradation efficiency for TOC and BPA.

##### Other Synthesis Methods

In addition to the methods presented in this section, there are other known methods for the synthesis of nanocomposite materials such as solvothermal methods. Solvothermal methods “offer a simple, direct, and low-temperature route to obtain nanometric particles with narrow size dispersions, and represent an alternative to calcinations for promoting crystallization under milder temperatures” [237]. It is mainly used for the synthesis of highly crystallized lanthanide (UCNPs) at relatively low temperature [238]. It has many advantages such as being a simple, economical and efficient method [239]. On the other hand, there are some disadvantages of this method such as the long processing time and the contraction that occurs during processing [240]. However, solvothermal method is very similar to the hydrothermal route, the only difference being that the precursor solution is usually non-aqueous [238]. Table 6 shows nanocomposite materials synthesized by other synthesis methods.

#### 2.2.2. Simulation of Nanocomposites

For the simulation of nanocomposites, VASP was the most used simulation software with 27 publications as shown in Figure 6. 10 of these publications were for the simulation of carbon-based material including CNTs and g-C_3_N_4_. However, 14 publications were for the simulation of other nanocomposites such as MnFe_2_O_4_ nanocubes using the PAW method, Fe_3_O_4_-HBPA-ASA using B3LYP functional group, and granular TiO_2_-La using the PBE method as shown in Table 7.

Figure 6 shows simulation software used to produce nanoparticles with the number of publications by each software.

As shown in Table 7, DFT (PBE), and DFT (PAW) have been mainly used for the simulation of the nanocomposite materials. This is due to the high accuracy of these two simulation methods. The augmented plane wave (PAW), and PBE was used to describe the electron–ion interactions [256]. For instance, Maji and colleagues used DFT (PBE) simulation method through VASP software for the simulation of Fe_2_O_3_-PC nanohybrids [203]. The results showed an agreement between the experimental and simulation results [203]. Similarly, Regmi and colleagues simulated an N-doped BiVO_4_ model using the DFT (PBE) method through VASP simulation software [208]. The results showed good agreement with experimental results in terms of the electronic property calculations such as the band structure and density of state (DOS) [208].

## 3. Water Filtration by Membrane Technology

Due to its important role in water purification, membrane technology is rapidly developing. The ability of rejection/adsorption may differ from one membrane to another due to the membrane pore size, surface charge, hydrophobicity/hydrophilicity, and surface morphology [257]. As shown in Figure 7, highly porous zeolitic imidazolate frameworks (ZIFs) have the highest adsorption rate for the removal of Uranium as ZIF-8 (540.4 mg/g) > Zn/Co-ZIF (527.5 mg/g) > ZIF-9 (448.6 mg/g) > ZIF-67 (368.2 mg/g). This is due to the high elimination capacity for Uranium because of (ZIF-8) large surface area and active metal ion [247]. However, porous graphitic carbon nitride (g-C_3_N_4_) has shown a high adsorption rate of Uranium with 149.7 mg/g. The results showed a strong interaction between uranyl and g-C_3_N_4_ (Ead = 156.83 kcal/mol) and the most effective sorption site was inside the holes of g-C_3_N_4_ [214]. Similarly, as shown in Figure 7, molybdenum disulphide (MoS_2_) showed a good elimination capacity of Uranium with 117.9, 45.7 and 37.1 (mg/g) [150,215]. This is due to the binding energy through U-S bond which improved by the molybdenum group [150].

In this section, we discuss the rejection rate and adsorption capacity of highly effective membranes including carbon-based membranes, metal oxides, and other nanocomposite membranes.

### 3.1. Carbon-Based Membranes

Carbon-based membranes including graphene, graphene oxide (GO), and carbon nanotubes (CNTs) are some of the most important nanocomposite membranes for the adsorption of pollutants from wastewater. Carbonaceous nanofiber (CNF), and graphene oxide have shown the highest adsorption rate for the removal of toxic metals of 341.2 and 288 mg/g, respectively, as shown in Figure 8. However, porous graphitic carbon nitride (g-C_3_N_4_), and carbon nanotubes (CNTs) have also shown a good adsorption rate for the removal of toxic metals such as Uranium and Indigo carmine (IC) dye of 149.7 and 136 mg/g, respectively. Figure 8 shows the rate of adsorption for toxic metals by carbon-based membranes. In addition, carbon-based membranes have been used for the removal of other toxic metals such as chloroform molecule (CHCl_3_), oils and organic solvents, and typical pharmaceuticals as shown in Table 8.

As shown in Table 8, carbon-based materials have been mainly used for the rejection/adsorption of toxic metals with high adsorption capacity (*qe*). Graphene oxide (GO) has been used for the removal of Ni(II), methylene blue (MB), arsenic (As(III)), 17 β- estradiol, and nitrate with high adsorption capacity ranging between 169–288 mg/g. However, carbon nanotubes (CNTs) have been used for the removal of Pb^2+^, sulfamethoxazole (SMZ) and ketoprofen (KET), bisphenol A, 17a-ethinyl estradiol (EE2), and typical pharmaceuticals, phosphate, indigo carmine (IC) dye, phenol separation, and binary CO_2_/N_2_ mixture. The rate of adsorption by carbon nanotubes (CNTs) was also high as it ranged between 0.64–136 mg/g. It can be observed that carbonaceous nanofiber/Ni-Al layered double hydroxide (CNF/LDH) showed an excellent adsorption capacity for the removal of Cu(II) and Cr(VI) with 219.6 and 341 mg/g, respectively.

### 3.2. Metal Oxides

Binary and ternary metal oxides have been widely studied for supercapacitors, and wastewater treatment because of its high structure stability, low cost, and high electronic conductivity [258,259]. Metal oxides including ZnO, and MnFe_2_O_4_ have shown a good adsorption capacity for the removal of barium ions and uranium with 64.4 and 119.9 mg/g, respectively. The high adsorption capacity of metal oxides is due to the ionic bonding which has important consequences for the adsorption of molecules at their surfaces [260]. In addition, Oct-Cu_2_O NCs showed an excellent adsorption capacity with 1112.6 mg/g of tetracycline on octahedral Cu_2_O nanocrystals. Table 9 shows rejection/adsorption capacity of metal oxide.

### 3.3. Other Nanocomposite Membranes

In addition to carbon-based materials and metal oxides, there are other nanocomposite membranes that can be used in the adsorption of pollutants from water. For instance, MoS_2_ has shown a high adsorption capacity for the removal of toxic metals such as lead (Pb^2+^) with 638 mg/g as shown in Figure 9. However, MOF-545, M-ATP, and Fe_3_O_4_-HBPA-ASA showed similar values of adsorption for Pb^2+^ as 73, 53.88, 88.36 mg/g, respectively. The high adsorption capacity of MoS_2_ is due to the strong ionic capture and electrostatic attractions which improve Pb^2+^ removal capacity under sunlight irradiation [151]. Table 10 shows rejection/adsorption capacity and the role for each nanocomposite membrane in the adsorption process.

As shown in Table 10, nanocomposite materials have shown a good rejection/adsorption capacity on toxic materials. For instance, Granular TiO_2_-La shows a good adsorption capacity of Arsenic (As III), and Fluoride (F) equal to 114 and 78.4 mg/g, respectively. Similarly, zirconium-based highly porous metal-organic framework (MOF-545) showed an excellent removal of Pb^2+^ with adsorption capacity up to 73 mg/g. In addition, the developed SPE-MOF-545 can be reused for up to 42 extraction cycles without a significant loss of extraction efficiency. Furthermore, Amino-modified attapulgite (M-ATP) also showed a high removal of Pb^2+^, and Cu^2+^ as follows: 53.58 and 28.86 mg/g, respectively.

## 4. Photocatalytic Degradation of Organic Pollutants

Photocatalytic degradation of organic pollutants has been widely used by researchers because of its great role in removing undesirable contaminants from water and wastewater [261]. The earliest mention of photocatalysis dates back to 1911, when German chemist Alexander Ebner incorporated the concept into his research on the illumination of zinc oxide (ZnO) on the bleaching of the dark blue dye, Prussian blue [262]. Around this time, Brunner and Kosack published an article discussing the degradation of oxalic acid in the presence of uranyl salts under illumination [263], while in 1913, Landau published an article explaining the phenomenon of photocatalysis [264]. Photo-Fenton oxidation is an advanced process uses hydroxyl radicals which increase the rate of degradation of organic pollutants [265]. While heterogeneous photocatalysis is one of the most studied processes for environmental purposes such as water purification and emission cleaning [266]. In this section, we discuss the role of titanium dioxide (TiO_2_), carbon nanomaterials, metal oxides, and other nanocomposites in the degradation of organic pollutants.

### 4.1. Titanium Dioxide (TiO_2_)

Titanium dioxide (TiO_2_) contributes significantly in the water purification process through degradation of organic pollutants such as Methylene blue (MB), Methyl orange, and Bisphenol A (BPA). In this process, TiO_2_ acts as a catalyst to accelerate photoreaction for the removal of organic pollutants. Table 11 shows the decomposition rate/degradation efficiency of titanium dioxide (TiO_2_).

As shown in Table 11, TiO_2_ has shown a high efficiency in the degradation of organic pollutants. For instance, Wang and colleagues synthesized C, N, F/TiO_2_NTs nanocomposite material for the degradation of methyl orange under UV-light and simulated sunlight [197]. The results showed a high photocatalytic activity under UV-light for C/TiO_2_NTs with degradation efficiency up to 100% [197]. Along the same lines, De Santiago et al. synthesized Cr-TiO_2_ nanocomposite supported on Fe_3_O_4_ for the degradation of malachite green dye (MG), and total organic carbon (TOC) [242]. The results showed a high removal under solar radiation up to 100% of MG, and 60% of TOC [242]. Furthermore, Cheng et al. synthesized titanate nanotubes supported on TiO_2_ (TiO_2_/TiNTs) for the removal of Cu(II), and phenanthrene under UV-light [200]. The results showed a high adsorption capacity of Cu(II) up to 115 mg/g, while for phenanthrene the degradation efficiency was more than 95% [200].

### 4.2. Carbon Nanomaterials

Carbon nanomaterials have showed an excellent contribution in the degradation of toxic materials with high efficiency for reduction of highly toxic contaminants up to 99%. As shown in Table 12, carbon nanomaterials have been used for the degradation of many toxic metals such as bisphenol A, Norfloxacin (NX), Methylene blue, and Tetracycline hydrochloride (TCH). For the removal of bisphenol A, Zhu and colleagues synthesized a heterogeneous Fenton catalyst (CNTs/Fh) [116], while Kim et al. synthesized ternary nanocomposites of Fe_3_O_4_ nanoparticles@ graphene–poly-N-phenylglycine nanofibers for the adsorption of Cu^2+^ with high degradation efficiency up to 95% [241]. Similarly, Shi et al. synthesized a CF/BiOBr/Ag_3_PO_4_ cloth for the degradation of tetracycline hydrochloride (TCH) up to 90% [154].

### 4.3. Metal Oxides

In addition to titanium dioxide (TiO_2_) and carbon nanomaterials, metal oxides can be used in the adsorption of pollutants from water. For instance, ZnO has shown a high adsorption capacity for the removal of toxic metals such as methylene blue, methyl orange dye (MO), and TOC with high degradation efficiency for TOC up to 80.4%. However, Cerium zirconium oxide (CexZryO_2_) also showed a high degradation efficiency for the removal of sulphonamides of 91.33% as shown in Table 13. Furthermore, Kitchamsetti and colleagues synthesized NiO nanobelt composite material for the removal of organic pollutants such as RhB, CV, MB, and MO. The results showed a high degradation efficiency up to 89%, 76.7%, 82.7%, and 79.1%, respectively [204].

### 4.4. Other Nanocomposites

For other nanocomposites, we demonstrate the decomposition rate/degradation efficiency for the most effective compounds for the degradation of organic pollutants such as nickel (Ni(II)), methylene blue (MB), sulfamethoxazole (SDZ), and trimethoprim (TMP). As shown in Table 14, for the degradation of MB, sodium titanate nanotubes (Na-TNT) showed a high degradation efficiency up to 99.5%, while Fe_2_O_3_-PC removed 75% of MB. For the degradation of sulfamethoxazole (SDZ), carbon dots/g-C_3_N_4_ (C-CN) heterostructures showed a high degradation efficiency up to 92.8% for the 3C-CN heterostructure. However, other nanocomposites such as black phosphorus quantum dots/Tubular g-C_3_N_4_ (BPQDs/TCN), CdSe-Ag-WO_3_-Ag photocatalyst, and 1D/2D W_18_O_49_/g-C_3_N_4_ nanocomposites, also showed a high degradation efficiency for the degradation of organic pollutants up to 96%.

## 5. Future Direction of Nanomembrane Adsorption Processes in Wastewater Treatment

In this review, we have discussed the role of the adsorption process in water and wastewater treatment through nanomembranes experimentally and by simulation. Below we will discuss the latest expected developments of the adsorption process in the near future with some recommendations.

With the rapid development of simulation software in some important DFT codes such as PBE, B3LYP and PAW, it will be easier to understand the physical and chemical properties of the adsorption process to fill the scientific gaps in realizing the adsorption mechanism, isotherm, kinetics, thermodynamics and other aspects of the adsorption process.Further economic feasibility studies should be conducted on adsorbents including the cost effectiveness of the choices of the materials, which is an important aspect of adsorption investigations.A huge improvement in the synthesis of nanomaterials using simulation will become possible by linking the density functional theory (DFT) codes using software such as Material Studio and Reactive forefield (ReaxFF) with the molecular dynamic (MD) simulation which will give more realism in acquiring accurate results before starting the experimental work. This step can reduce costs of conducting trials and save time.Evolution in the ability and durability of nanomembranes in selectivity of undesirable materials by adsorption which increase the adsorption capacity (*qe*). This is possible by improving the mechanical properties of the nanomembranes by creating special nanocomposites such as graphene/TiO_2_, and graphene/MoS_2_. These two nanocomposites have proven their ability to expel salt and permeate water with high efficiency, so we expect a high adsorption capacity (*qe*) from them.More comprehensive studies should be conducted on the effect of multi-layer membranes in the adsorption process, which is expected to increase the adsorption capacity (*qe*) due to the increase in attractions between organic pollutants and membranes. In addition, it is possible to use different layers in the same system which can adsorb different pollutants at the same time. We recommend simulating the system using a molecular dynamic simulation software using two different layers and then testing the possibility of adsorption on different organic pollutants.More studies should be conducted on the possibility of developing more effective forcefields which is highly required in some molecular dynamics simulation software. Creating and developing high effective forcefields will increase the possibility of simulating all kinds of atoms and molecules with high accuracy without errors.

## 6. Conclusions

Based on the unique properties of the adsorption technique and its excellent contribution in water and wastewater treatment, it has attracted researchers with great interest in the membrane-based separation field. In this review paper, the preparation of nanomembranes and nanocomposite materials were summarized by experimental and simulation methods. Then we focused on water treatment by membrane technology using the adsorption technique. The main characteristics related to adsorption technology are discussed including the adsorbed/rejected material, rate of rejection/adsorption capacity, and the role of the nanocomposite membrane. In synthesis of nanocomposite materials, we have discussed the latest development techniques such as the hydrothermal method, chemical vapor deposition (CVD), and one-pot synthesis. In this review, the hydrothermal method was the most common method used as it contributed in 56% of the total papers reviewed, while chemical vapor deposition and one-pot synthesis methods contributed 7% and 6%, respectively. This is because of the hydrothermal method’s low-cost synthesis, being simple and easy. Graphene oxide (GO) showed promising results in water and wastewater treatment by using the adsorption technique. GO membrane has showed a high adsorption capacity up to 288 mg/g for the removal of As(III), while for the removal of Ni (II), the adsorption capacity equalled 197.8 mg/g. In addition, GO has been used for the removal of 17 β- estradiol with high adsorption capacity up to 169.49 mg/g. In addition, GO contributed to the degradation of organic pollutants such as methylene blue (MB) with a degradation efficiency up to 99.3%. In addition to GO, metal oxides and other nanocomposites has also showed a high adsorption capacity for the removal of toxic metals such as lead (Pb^2+^), barium ions and uranium. Highly porous zeolitic imidazolate frameworks (ZIFs) has the highest adsorption rate for the removal of Uranium of 540.4 mg/g. For the removal of bisphenol A, PyTTA-Dva-COF membrane had the highest adsorption capacity of 285 mg/g. For the degradation of organic pollutants, TiO_2_ nanoflowers showed an excellent contribution for the degradation of Bisphenol A (BPA), Diphenyl phenol, P-tert-butyl phenol, and Resorcinol under UV-light up to 95%. Moreover, Cr-TiO_2_ degraded 100% of Malachite green dye (MG), and 60% of total organic carbon (TOC). In a comparison with other nanocomposite materials, metal oxides including ZnO@C, Cerium zirconium oxide (CexZryO_2_), ZnO/Al_2_O_3_, and NiO nanobelt have also showed good results for the degradation of MB, sulphonamides up to 91.33%, TOC, and other organic pollutants. Finally, special attention has been paid to the future direction of nanomembrane adsorption processes in water and wastewater treatment. The need for an economic feasibility study on adsorbents was mainly highlighted with the need to connect DFT with MD simulation for more realism before starting the experimental work.

## Figures and Tables

**Figure 1 membranes-12-00360-f001:**
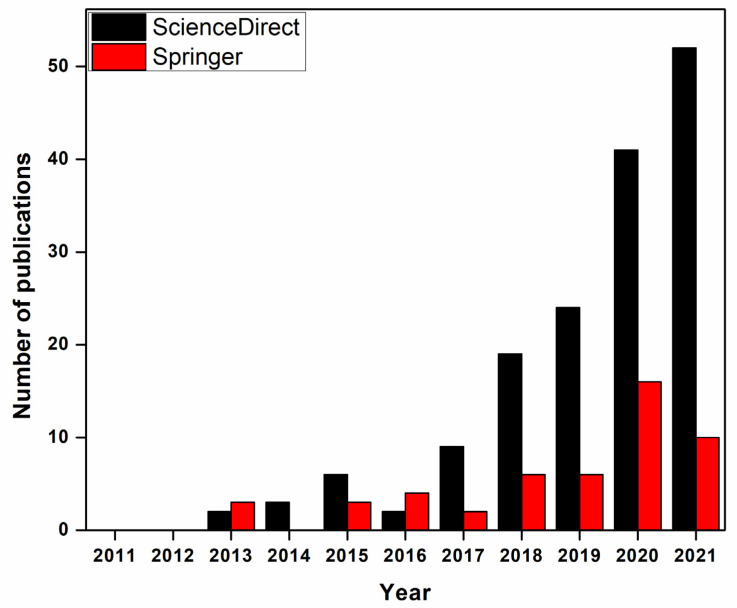
Number of publications in water and wastewater treatment by adsorption (experimentally validated by simulation) between 2011 and 2021 (topic keywords “adsorption”, “nanomaterials” “DFT” “simulation” and “wastewater treatment” searched from ScienceDirect and Springer), data updated 16 June 2021.

**Figure 2 membranes-12-00360-f002:**
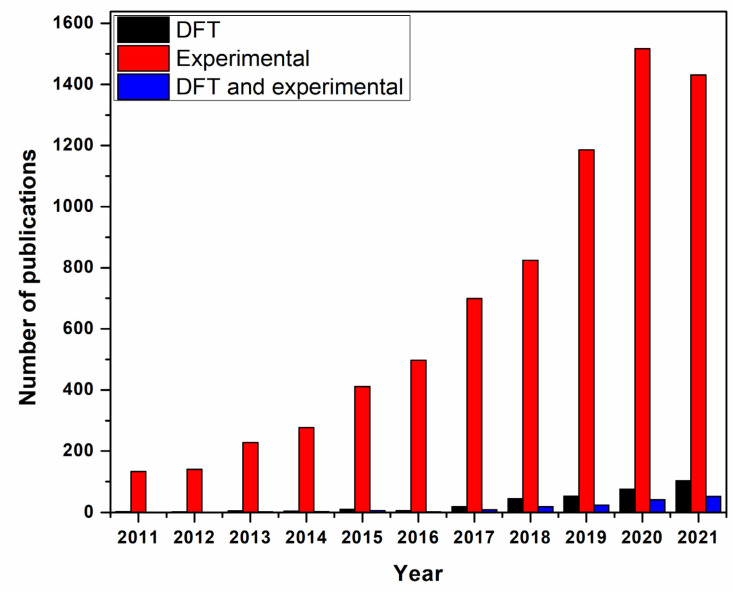
Number of publications in water and wastewater treatment by adsorption (experimentally, simulation (DFT), and experimentally validated by simulation) between 2011 and 2021. Data updated 23 June 2021 using ScienceDirect database.

**Figure 3 membranes-12-00360-f003:**
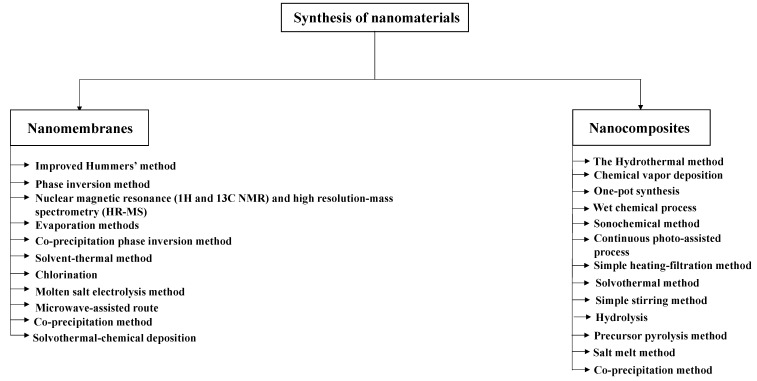
The most common methods used in the synthesis of nanomaterials.

**Figure 4 membranes-12-00360-f004:**
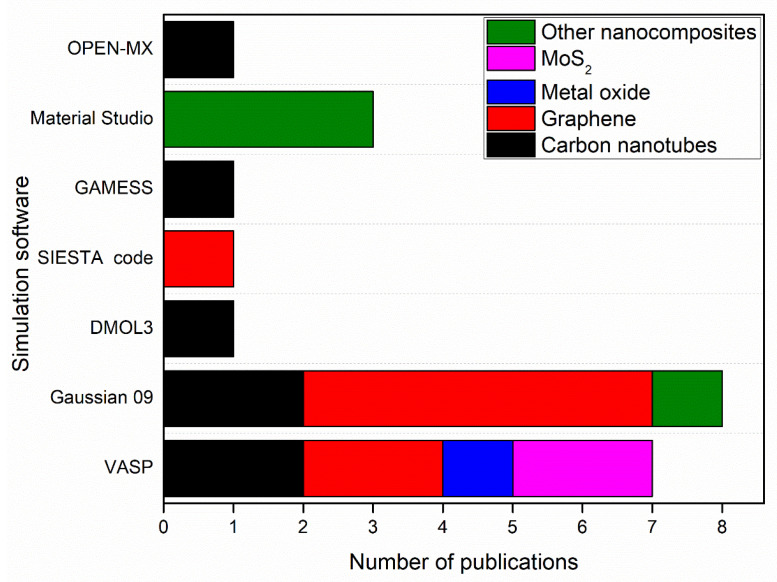
Simulation software used to produce nanomembranes with the number of publications by each software.

**Figure 5 membranes-12-00360-f005:**
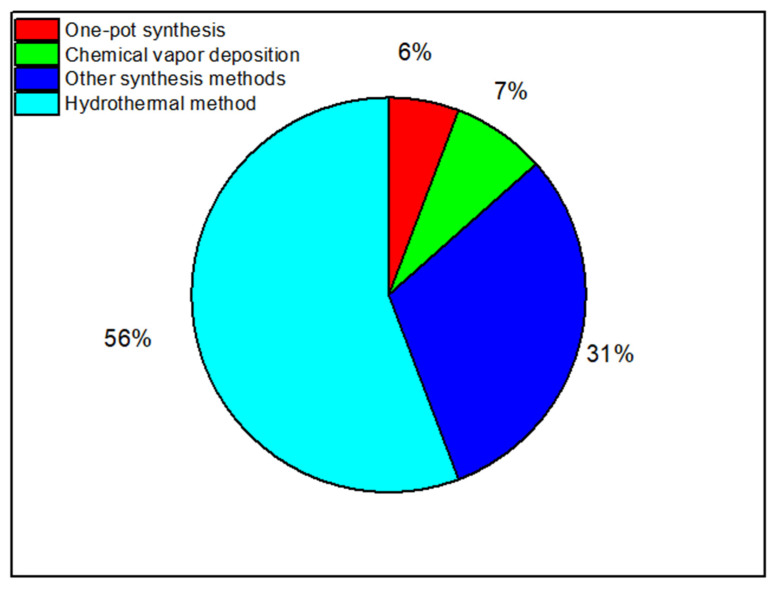
The percentages of the number of publications by each experimental method used in the synthesis of nanocomposite materials.

**Figure 6 membranes-12-00360-f006:**
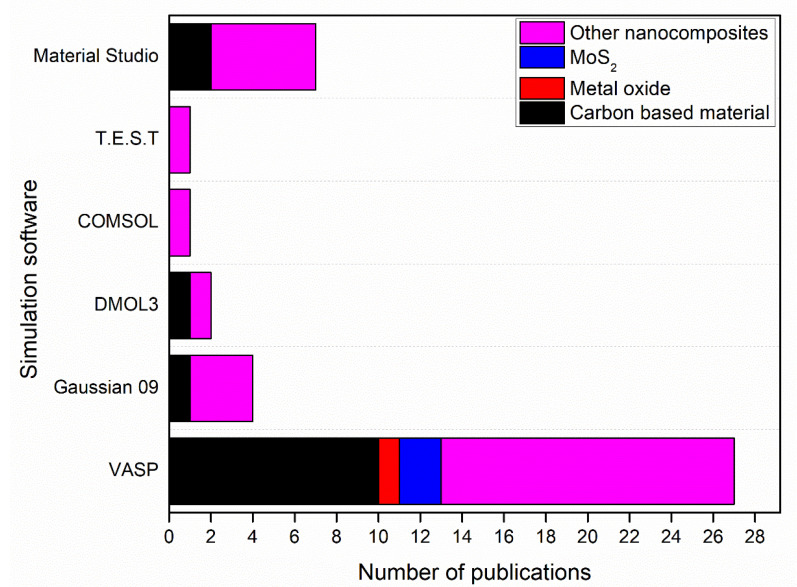
Simulation software used to produce nanoparticles with the number of publications by each software.

**Figure 7 membranes-12-00360-f007:**
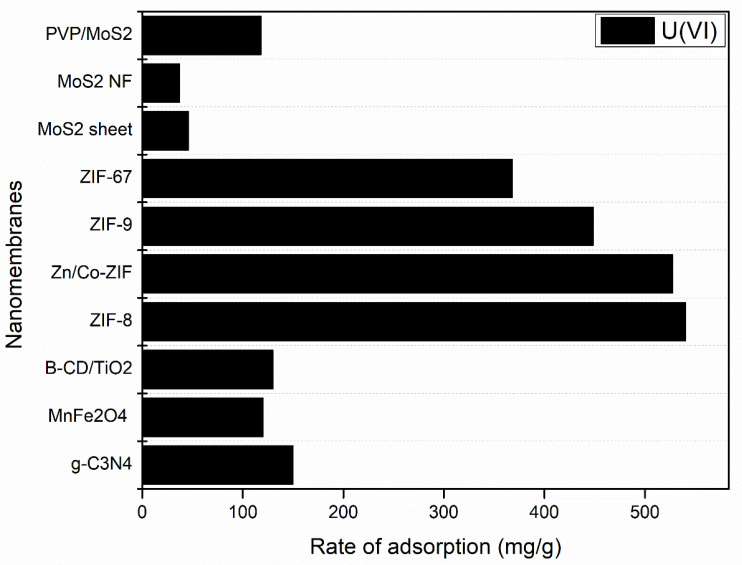
Rate of adsorption for different nanomembranes for the removal of Uranium.

**Figure 8 membranes-12-00360-f008:**
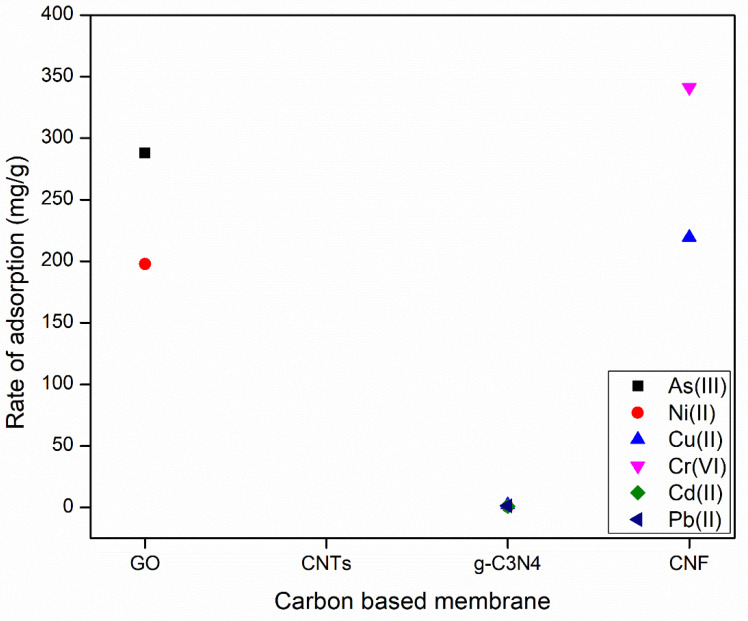
Rate of adsorption for toxic metals by carbon-based membranes. The adsorption rates are difficult to distinguish in the Figure for g-C_3_N_4_ for Cd(II), Cu (II), Ni(II) and Pb(II) which equal 1.00, 2.09, 0.64, and 1.36 mmol/g, respectively.

**Figure 9 membranes-12-00360-f009:**
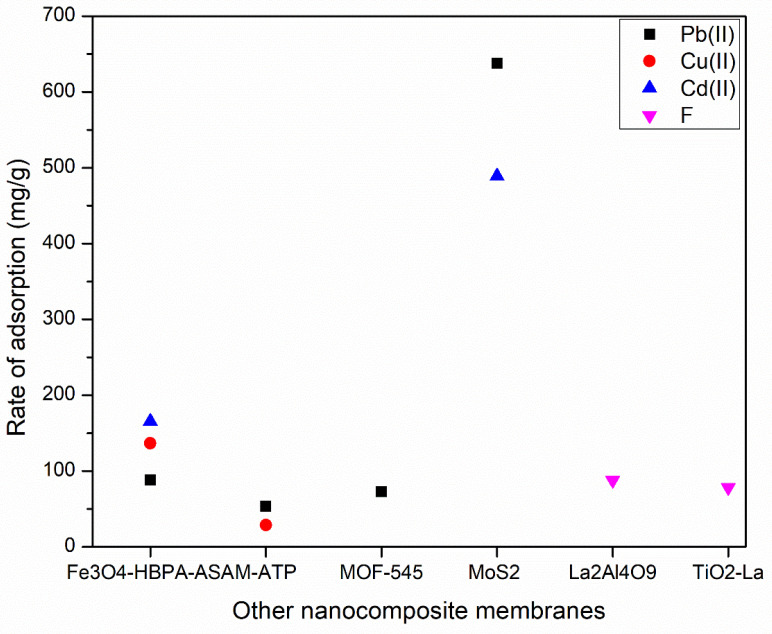
Rate of adsorption for Pb^2+^ by other nanocomposite membranes.

**Table 1 membranes-12-00360-t001:** Nanomembranes synthesized by different synthesis methods.

Membrane	Material Type	Synthesis Method	Reference
SWCNTs	Carbon nanotube (CNT)	Obtained from Cheap Tubes, Inc.	[137]
Graphene oxide	Oxidized graphene oxide	Obtained commercially from Sigma Aldrich	[113]
ZnO surface	Zinc oxide (ZnO)	Evaporation methods	[138]
MnFe_2_O_4_ nanocubes	Manganese ferrite nanoparticles (MnFe_2_O_4_)	Co-precipitation phase inversion method	[139]
Graphene	3D foam graphene	Obtained commercially	[140]
MGOA	Graphene oxide (GO), ammonium (NH_4_^+^)	Modified Hummers’ method	[141]
PyTTA-Dva-COF	Nitrogen (N), covalent organic framework	Solvent-thermal method	[142]
Ultrafiltration PSF/GO membrane	Graphene oxide (GO), polysulfone (PSF)	Phase inversion method	[143]
Nitrogen doped carbon (CNs)	Carbon (C), nitrogen (N), titanium (Ti)	Chlorination	[144]
Graphene oxide	Graphene oxide	Improved Hummers’ method	[145]
Single-layer graphene nanosheets	Graphite	Solution-phase exfoliation integrating bath sonication and microwave irradiation in organic solvents	[146]
Carbon nanotubes (CNTs)	Carbon nanotube (CNT)	Nuclear magnetic resonance (1H and 13C NMR) and high resolution-mass spectrometry (HR-MS)	[147]
Graphene oxide	Graphene oxide	Modified Hummers’ method	[148]
Graphene oxide	Graphene oxide	Modified Hummers’ method	[149]
MoS_2_ nanosheets	Molybdenum disulphide	Molten salt electrolysis method	[150]
MoS_2_ nanosheets	Molybdenum disulphide	Microwave-assisted route	[151]
Zn–Fe LDH	Zinc (Zn), iron (Fe)	Co-precipitation method	[152]
Lanthanum-aluminium perovskite (La_2_Al_4_O_9_)	Lanthanum (La), aluminium (Al)	Obtained commercially from Aladdin company	[153]
CF/BiOBr/Ag_3_PO_4_ cloth	Carbon fibre (CF), bismuth oxybromide (BiOBr), silver phosphate (Ag_3_PO_4_)	Solvothermal-chemical deposition	[154]

**Table 2 membranes-12-00360-t002:** Nanomembranes simulated by different simulation software.

Membrane	Software	Simulation Method	Mathematical Model	Reference
(O-CNTs), (G-CNTs)	Gaussian 09W	DFT (B3LYP functional group)	Integral Equation Formalism Polarized Continuum Model (IEFPCM)	[163]
Graphene	VASP	DFT (PAW)	Kohn-Sham equations	[140]
Graphene oxide	SIESTA code	DFT (LDA)	Kohn-Sham equations	[113]
MGOA	Gaussian 09	DFT (B3LYP functional group)	Thomas, Yoon–Nelson, and Adams–Bohart models	[141]
PyTTA-Dva-COF	Gaussian 09	DFT (B3LYP functional group)	ONIOM model	[142]
Vertically aligned (VA) CNT (open-end) hybrid membrane	DMOL3 package	DFT (PW91)	Exchange-Correlation functional	[164]
Ultrafiltration PSF/GO membrane	OPEN-MX software	DFT (LDA)	Hoffmann’s model	[143]
Graphene oxide	Gaussian 09	DFT (Gaussian-Lorentzian function)	Exchange-Correlation functional	[145]
S, N co-doped graphene aerogel (SN-rGO-A)	Gaussian 09	DFT (B3LYP functional group)	Thomas, Yoon–Nelson, and Adams–Bohart models	[165]
ZIF8@carbon nanotube	VASP	DFT (PBE)	Exchange-Correlation functional	[166]
Carbonaceous nanofiber/Ni-Al layered double hydroxide (CNF/LDH)	VASP	DFT (PAW)	Kohn-Sham equations	[167]
SWCNTs, MWCNTs, and PAC	GAMESS	DFT (B3LYP5 functional)	Exchange-Correlation functional	[168]
Single-layer graphene nanosheets	VASP	DFT (PAW)	Kohn-Sham equations	[146]
Graphene oxide	Gaussian 09	DFT (PBE1PBE functional model)	Exchange-Correlation functional	[148]
Graphene oxide	Gaussian 09	DFT (B3LYP/6-31G* level)	Exchange-Correlation functional	[149]
ZnO surface	VASP	DFT (PBE)	Exchange-Correlation functional	[138]
MoS_2_ nanosheets	VASP	DFT (PAW)	Kohn-Sham equations	[150]
Zn–Fe LDH	Materials Studio (BIOVIA, 2017)	DFT (DMol^3^) code	Exchange-Correlation functional	[152]
Lanthanum-aluminium perovskite (La_2_Al_4_O_9_)	Materials Studio	DFT (PBE)	Exchange-Correlation functional	[153]
MoS_2_ nanosheets	VASP	DFT (PAW)	Kohn-Sham equations	[151]
SWCNTs	Gaussview	DFT (B3LYP5) functional	Exchange-Correlation functional	[137]
CF/BiOBr/Ag_3_PO_4_ cloth	Materials Studio	DFT (GGA-PBE)	Exchange-Correlation functional	[154]

**Table 3 membranes-12-00360-t003:** Nanocomposite materials synthesized by the hydrothermal method.

Nanocomposite Material	Material Type	Reference
Heterogeneous Fenton catalysts (CNTs/Fh)	Oxidized carbon nanotubes (CNTs), ferrihydrite (Fh)	[116]
(N-rGO/BiVO_4_)	Bismuth vanadate (BiVO_4_), reduced graphene oxide (rGO), nitrogen (N)	[193]
ZnO@C	Zinc Oxide (ZnO), carbon (C)	[194]
Cerium zirconium oxide (CexZryO_2_)	Cerium (Ce), zirconium oxide (ZrO_2_)	[195]
ZnO/Al_2_O_3_	Zinc oxide (ZnO), aluminium oxide (Al_2_O_3_).	[196]
C, N, F/TiO_2_NTs	Carbon (C), nitrogen (N), fluoride (F), titanium dioxide nanotubes (TiO_2_NTs)	[197]
iN-Ti_3_C_2_/TiO_2_ hybrid	Titanium carbide (Ti_3_C_2_), titanium dioxide (TiO_2_), isopropyl amine, nitrogen (N)	[198]
TiO_2_ nanoflowers (TNFs)	Titanium dioxide (TiO_2_)	[199]
Titanate nanotubes supported TiO_2_ (TiO_2_/TiNTs)	Titanium dioxide (TiO_2_), titanate nanotubes	[200]
Black phosphorus quantum dots/Tubular g-C_3_N_4_ (BPQDs/TCN)	Black phosphorus (BP), tubular g-C_3_N_4_	[201]
Sodium titanate nanotubes (Na-TNT)	Sodium (Na), titanate nanotubes (TNT)	[202]
Fe_2_O_3_-PC nanohybrids	Iron oxide (Fe_2_O_3_)	[203]
NiO nanobelt	Nickel oxide (NiO)	[204]
Carbon dots/g-C_3_N_4_ (C-CN) heterostructures	Graphitic Carbon Nitride (g-C_3_N_4_)	[205]
AgBr/h-MoO_3_	Silver bromide (AgBr), hexagonal molybdenum oxide (h-MoO_3_)	[206]
Hybrid catalysts (CN-CGs)	Coal gangue (CG), graphitic carbon nitride g-C_3_N_4_ (CN)	[207]
N-doped BiVO_4_	Nitrogen (N), bismuth vanadate (BiVO_4_)	[208]
PPECu thin film electrode	Copper (Cu), phenylacetylene (PPE)	[209]
Fe_x_Mo_1-x_S_2_ catalysts	Iron (Fe), Molybdenum disulfide (MoS_2_)	[210]
P-doped porous g-C_3_N_4_	Graphitic carbon nitride (g-C_3_N_4_), phosphorus (P)	[211]
1D/2D W_18_O_49_/g-C_3_N_4_ nanocomposites	Graphitic carbon nitride (g-C_3_N_4_), oxygen-deficient tungsten oxide (W_18_O_49_)	[212]
Oct-Cu_2_O NCs	Cuprous oxide (Cu_2_O)	[213]
g-C_3_N_4_	Graphitic carbon nitride (g-C_3_N_4_)	[214]
ZIF8@carbon nanotube	Carbon nanotube (CNT), zeolitic imidazole framework-8 (ZIF8)	[166]
CNF/LDH	Carbonaceous nanofiber (CNF), nickel (Ni), aluminium (Al)	[167]
PVP/MoS_2_	Molybdenum disulphide, polyvinylpyrrolidone	[215]
β-CD/TiO_2_	Titanium dioxide (TiO_2_), β-cyclodextrin C_42_H_70_O_35_	[216]
MOF-545	Zirconyl chloride octahydrate, Sigma-Aldrich; porphyrin, H4-Tcpp-H2, TCl	[217]

**Table 4 membranes-12-00360-t004:** Nanocomposite materials synthesized by the chemical vapor deposition (CVD) method.

Nanocomposite Material	Material Type	Reference
Co_3_O_4_/CNTs	Carbon nanotubes (CNTs), cobalt tetra-oxide (Co_3_O_4_)	[231]
O-CNTs, G-CNTs	Oxidized carbon nanotubes (O-CNTs), graphitized carbon nanotubes (G-CNTs).	[163]
Vertically aligned (VA) CNT (open-end) hybrid membrane	Carbon nanotube (CNT), polydimethylsiloxane (PDMS) membrane	[164]
COOH/CNTs	Carbon nanotubes (CNTs), carboxylic functionalized groups (COOH)	[232]

**Table 5 membranes-12-00360-t005:** Nanocomposite materials synthesized by the one-pot synthesis method.

Nanocomposite Material	Material Type	Reference
S, N co-doped graphene aerogel (SN-rGO-A)	Graphene oxide (GO), sulfur (S), nitrogen (N).	[165]
ZIF-67 Carbocatalysts, Nitrogen-doped magnetic carbon (Co@N-C)	Cobalt (Co), nitrogen (N), carbon (C)	[235]
Fe/Fe_3_C@PC	Graphitized porous carbon (PC), Fe-based nanoparticle core (Fe/Fe_3_C)	[236]

**Table 6 membranes-12-00360-t006:** Nanocomposite materials synthesized by other synthesis methods.

Nanocomposite Material	Material Type	Synthesis Method	Reference
Ternary nanocomposites of Fe_3_O_4_ nanoparticles@ graphene–poly-N-phenylglycine nanofibers	Graphene oxide (GO), nitrogen (N), iron oxide (Fe_3_O_4_), phenylglycine (C_6_H_5_CHCO_2_H).	Wet chemical process	[241]
Cr-TiO_2_ supported on Fe_3_O_4_	Titanium dioxide (TiO_2_), chromium (Cr), iron oxide black (Fe_3_O_4_).	Sonochemical method	[242]
CdSe-Ag-WO_3_-Ag photocatalyst	Cadmium selenide (CdSe), silver (Ag), tungsten trioxide (WO_3_).	Continuous photo-assisted process	[243]
Bi/Fe0	Bismuth (Bi), iron (Fe)	Simple chemical reactions	[244]
Granular carbon nanotubes (CNTs)	Carbon nanotubes (CNTs)	Simple heating-filtration method	[245]
SWCNTs, MWCNTs, and PAC	Carbon nanotubes (CNTs)	SWCNTs: Obtained commercially from Cheap Tubes, Inc. MWCNTs: Obtained commercially from Sigma Aldrich.	[168]
Fe_3_O_4_-HBPA-ASA	Magnetite (Fe_3_O_4_)	Solvothermal method	[246]
Highly porous zeolitic imidazolate frameworks (ZIFs)	Highly porous zeolitic imidazolate frameworks	Simple stirring method	[247]
Granular TiO_2_-La	Titanium dioxide (TiO_2_), lanthanum (La)	Hydrolysis	[248]
Ni (II) modified porous BN	Nickel (Ni), boron nitride (BN)	Precursor pyrolysis method	[249]
Bi_2_O_2_CO_3_ nanosheets	Bismuth carbonate	Simple stirring method	[250]
Amino-modified attapulgite (M-ATP)	Attapulgite clay, the 3-aminopropyltriethoxysilane, Pb (NO_3_)_2_ and Cu(NO_3_)_2_	Simple stirring method	[251]
g-C_3_N_4_	Graphitic carbon nitride (g-C_3_N_4_)	Salt melt method	[252]
MIL-101(Fe) and MIL-101(Fe,Co)	MIL-101(Fe)	Solvothermal method	[253]
CuCo_2_O_4_/BiVO_4_	Bismuth vanadate (BiVO_4_)	Solvothermal method	[254]
Zn/Fe LDH	Zinc (Zn), iron (Fe)	Co-precipitation method	[255]

**Table 7 membranes-12-00360-t007:** Nanocomposite materials simulated by different simulation software.

Nanocomposite Material	Software	Simulation Method	Mathematical Model	Reference
Nitrogen doped carbon (CNs)	VASP	DFT (PAW)	Kohn-Sham equations	[144]
COOH/CNTs	DMol3 program	DFT (PBE)	Exchange-Correlation functional	[232]
Porous graphitic carbon nitride (g-C_3_N_4_)	VASP	DFT (PAW)	Kohn-Sham equations	[214]
Granular carbon nanotubes (CNTs)	Not supplied	DFT	The Langmuir model	[245]
Carbon nanotubes (CNTs)	Gaussian 09	DFT (Minnesota dispersion functional, M06-2×/6–31G(d) level)	Exchange-Correlation functional	[147]
MnFe_2_O_4_ nanocubes	VASP	DFT (PAW)	Kohn-Sham equations	[139]
Oct-Cu_2_O NCs	VASP	DFT (PW91)	Exchange-Correlation functional	[213]
Amino-modified attapulgite (M-ATP)	VASP	DFT (PBE)	Exchange-Correlation functional	[251]
β-CD/TiO_2_	VASP	DFT (PAW)	Kohn-Sham equations	[216]
Fe_3_O_4_-HBPA-ASA	Gaussian 16 package	DFT (B3LYP)	Exchange-Correlation functional	[246]
PVP/MoS_2_	VASP	DFT (PAW)	Kohn-Sham equations	[215]
Highly porous zeolitic imidazolate frameworks (ZIFs)	Gaussian 09	DFT (B3LYP)	Exchange-Correlation functional	[247]
Ni (II) modified porous BN	VASP	DFT (PAW)	Kohn-Sham equations	[249]
CuCo_2_O_4_/BiVO_4_	Materials Studio 6.0 (2011)	DFT (PBE)	Exchange-Correlation functional	[254]
Granular TiO_2_-La	Materials Studio 7.0	DFT (PBE)	Exchange-Correlation functional	[248]
g-C_3_N_4_	Not supplied	DFT	Langmuir model, and Freundlich model	[252]
MOF-545	Not supplied	DFT	Exchange-Correlation functional	[217]
MIL-101(Fe) and MIL-101(Fe, Co)	DMol3 code	DFT (PBE)	Exchange-Correlation functional	[253]
Bi_2_O_2_CO_3_ nanosheets	VASP 5.4	DFT (HSE06)	Exchange-Correlation functional	[250]
Zn/Fe LDH	Materials Studio (BIOVIA, 2017)	DFT (GGA-RPBE)	Exchange-Correlation functional	[255]
Heterogeneous Fenton catalysts (CNTs/Fh)	VASP	DFT (PAW)	Kohn-Sham equations	[116]
Co_3_O_4_/CNTs	Material studio 2017	DFT (PBE)	Exchange-Correlation functional	[231]
(ZIF-67 Carbocatalysts), Nitrogen-doped magnetic carbon (Co@N-C)	VASP	DFT (PBE)	Exchange-Correlation functional	[235]
ternary nanocomposites of Fe_3_O_4_ nanoparticles@ graphene–poly-N-phenylglycine nanofibers	VASP	DFT (RPBE)	Exchange-Correlation functional	[241]
(N-rGO/BiVO_4_)	Not supplied	DFT	Exchange-Correlation functional	[193]
Cerium zirconium oxide (CexZryO_2_)	VASP	DFT (PBE)	Exchange-Correlation functional	[195]
NiO nanobelt	VASP	DFT (PAW)	Kohn-Sham equations	[204]
ZnO/Al_2_O_3_	VASP, COMSOL	DFT (PBE)	Exchange-Correlation functional	[196]
ZnO@C	Molecular Operating Environment software (MOE, 2008.10)	DFT	Exchange-Correlation functional	[194]
C, N, F/TiO_2_NTs	VASP	DFT	Exchange-Correlation functional	[197]
iN-Ti_3_C_2_/TiO_2_ hybrid	VASP	DFT (PBE)	Exchange-Correlation functional	[198]
TiO_2_ nanoflowers (TNFs)	VASP	DFT (PBE)	Exchange-Correlation functional	[199]
Cr-TiO_2_ supported on Fe_3_O_4_	Not supplied	DFT (M06 L)	a Langmuir-Hinshelwood model	[242]
Titanate nanotubes supported TiO_2_ (TiO_2_/TiNTs)	Gaussian 03	DFT (B3LYP)	Exchange-Correlation functional	[200]
Black phosphorus quantum dots/Tubular g-C_3_N_4_ (BPQDs/TCN)	Materials Studio	DFT (PBE)	Exchange-Correlation functional	[201]
CdSe-Ag-WO_3_-Ag photocatalyst	VASP	DFT (PBE)	Exchange-Correlation functional	[243]
Sodium titanate nanotubes (Na-TNT)	Materials Studio	DFT (RPBE)	Exchange-Correlation functional	[202]
Fe_2_O_3_-PC nanohybrids	VASP	DFT (PBE)	Exchange-Correlation functional	[203]
Carbon dots/g-C_3_N_4_ (C-CN) heterostructures	VASP	DFT (PBE)	Exchange-Correlation functional	[205]
AgBr/h-MoO_3_	Toxicity Estimation Software Tool (T.E.S.T.)	DFT (QSAR)	Exchange-Correlation functional	[206]
Hybrid catalysts (CN-CGs)	VASP	DFT (GGA-PBE)	Exchange-Correlation functional	[207]
Fe/Fe_3_C@PC	VASP, Version 5.4.1	DFT (PAW)	Kohn-Sham equations	[236]
N-doped BiVO_4_	VASP	DFT (PBE)	Exchange-Correlation functional	[208]
Bi/Fe0	Materials Studio	DFT (PBE)	Exchange-Correlation functional	[244]
PPECu thin film electrode	VASP	DFT (PAW)	Kohn-Sham equations	[209]
Fe_x_Mo_1-x_S_2_ catalysts	VASP	DFT (PBE)	Exchange-Correlation functional	[210]
P-doped porous g-C_3_N_4_	VASP	DFT (PBE)	Exchange-Correlation functional	[211]
1D/2D W_18_O_49_/g-C_3_N_4_ nanocomposites	VASP	DFT (PAW)	Kohn-Sham equations	[212]

**Table 8 membranes-12-00360-t008:** Rejection/adsorption capacity of carbon-based membranes.

Membrane	Role of Carbon-Based Membrane	Rejected/Adsorbed Material	Rate of Rejection (%)/Adsorption Capacity (*qe*) (mg/g)	Reference
O-CNTs, G-CNTs	Adsorption of Pb^2+^ on O-CNTs and G-CNTs	Pb^2+^	<9.03%	[163]
Vertically aligned (VA) CNT (open-end) hybrid membrane	Gas separation	Phenol separation binary CO_2_/N_2_ mixture separation	Not supplied	[164]
COOH/CNTs	Adsorptive removal of Indigo carmine (IC) dye onto nanotube carbon (CNTs)	Indigo carmine (IC) dye	CNT: (88.5 mg/g) COOH-CNT: (136 mg/g)	[232]
Granular carbon nanotubes (CNTs)	Efficient removal of typical pharmaceuticals	Typical pharmaceuticals	CBZ: 0.3695 mg/g TC: 0.2842 mg/g DS: 0.2031 mg/g	[245]
ZIF8@carbon nanotube	Adsorption of Phosphate on ZIF-8@MWCNT	Phosphate	(92.8–100%)	[166]
SWCNTs, MWCNTs, and PAC	Adsorption of bisphenol A and 17a-ethinyl estradiol (EE2) using carbon nanomaterials and powdered activated carbon	bisphenol A, 17a-ethinyl estradiol (EE2)	90% removal of both BPA and EE2	[168]
Carbon nanotubes (CNTs)	Adsorption of Sulfamethoxazole (SMZ) and ketoprofen (KET) on modified carbon nanotubes (CNTs)	Sulfamethoxazole (SMZ) and ketoprofen (KET)	Adsorption percentage: SMZ: >70% KET >80% Removal percentage: SMZ: 30% KET: >50%	[147]
Graphene	Adsorption of CHCl_3_ on graphene	Chloroform molecule (CHCl_3_)	Not supplied	[140]
Single-layer graphene nanosheets	Desalination and ion capture by sunlight single layer graphene nanosheet	Na^+^, Pb^2+^ and Fe^3+^	Na^+^: 86.1% Pb^2+^: 77.3% Fe^3+^: 46.1%	[146]
Graphene oxide	Adsorption of 17 β- estradiol on graphene oxide	17 β- estradiol	169.49 mg/g	[113]
Graphene oxide	Adsorption of As(III) on graphene oxide	As(III)	288 mg/g	[145]
Graphene oxide	Removal of Ni(II) from wastewater by adsorption on graphene oxide surface	Ni(II)	197.8 mg/g	[148]
Graphene oxide	Adsorption of Methylene blue (MB) on graphene oxide surface	Methylene blue (MB)	Not supplied	[149]
MGOA	Adsorption of quinoline in wastewater	Quinoline pollutants	103 mg/g	[141]
Ultrafiltration PSF/GO membrane	Nitrate rejection, antifouling property	Nitrate	22.5% at 0.5 weight percent of GO	[143]
SN-rGO-A	Adsorb oils and organic solvents by SN-rGO-A	Oils and organic solvents	*qe*: 65–192 times its weight	[165]
Nitrogen doped carbon (CNs)	Adsorbent for the removal of anionic heavy metals from wastewater and sewage	Arsenic	31.08 mg/g	[144]
g-C_3_N_4_	Adsorptive removal of uranyl by porous graphitic carbon nitride (g-C_3_N_4_)	Uranium	149.70 mg/g	[214]
g-C_3_N_4_	Removal of heavy metal ions from aqueous solutions	Pb(II), Cu(II), Cd(II) and Ni(II))	Pb(II): 1.36 mmol/g Cu(II): 2.09 mmol/g Cd(II): 1.00 mmol/g Ni(II): 0.64 mmol/g	[252]
Carbonaceous nanofiber/Ni-Al layered double hydroxide (CNF/LDH)	Removal of heavy metals from aqueous solutions	Cu(II), Cr(VI)	Cu(II): 219.6 mg/g Cr(VI): 341.2 mg/g	[167]

**Table 9 membranes-12-00360-t009:** Rejection/adsorption capacity of metal oxides.

Membrane	Role of Metal Oxide	Rejected Material	Adsorption Capacity (*qe*) (mg/g)	Reference
ZnO surface	Removal of barium (Ba^2+^) ions on ZnO spherical nanoparticles	Barium ions	64.6 mg/g	[138]
MnFe_2_O_4_ nanocubes	High adsorption capacity of U(VI) and Eu(III) on magnetic MnFe2O4 nanocubes	Uranium U(VI) Eu(III)	U(VI): 119.90 mg/g Eu(III): 473.93 mg/g	[139]
Oct-Cu_2_O NCs	Adsorption of tetracycline on octahedral Cu_2_O nanocrystals	Tetracycline	1112.6 mg/g	[213]

**Table 10 membranes-12-00360-t010:** Rejection/adsorption capacity of other nanocomposite membranes.

Membrane	Role of Nanocomposite Membrane	Rejected Material	Rate of Rejection (%)/Adsorption Capacity (*qe*) (mg/g)	Reference
PyTTA-Dva-COF	Removal of bisphenol A from aqueous solution	bisphenol A	285 mg/g	[142]
Zn–Fe LDH	Removal of diclofenac from water using Zn–Fe LDH	Diclofenac	74.50 mg/g	[152]
Lanthanum-aluminium perovskite (La_2_Al_4_O_9_)	Adsorption mechanisms for removing fluoride using lanthanum-aluminum perovskite	Fluoride (F)	87.75 mg/g	[153]
β-CD/TiO_2_	Adsorption mechanisms for uranium removal by β-CD/TiO_2_	U(VI)	129.8 mg/g	[216]
Fe_3_O_4_-HBPA-ASA	Removal of heavy metal ions from aqueous solution by Fe3O4-HBPA-ASA	Heavy metal ions	Cu(II): 136.66 mg/g Pb(II): 88.36 mg/g Cd(II): 165.46 mg/g	[246]
ZIFs	Highly efficient removal of U(IV)	U(VI)	ZIF-8: 540.4 mg/g Zn/Co-ZIF: 527.5 mg/g ZIF-9: 448.6 mg/g ZIF-67: 368.2 mg/g	[247]
Granular TiO_2_-La	Adsorption of arsenic and fluoride using granular TiO_2_-La	Arsenic (As III), fluoride (F)	As(III): 114 mg/g F: 78.4 mg/g	[248]
Ni (II) modified porous BN	Removal of tetracycline from aqueous solution	Tetracycline (Tc)	429.582 mg/g Removal percentage: 99.769%	[249]
Bi_2_O_2_CO_3_ (BOC) nanosheets with oxygen vacancies	Removal of (NO) by BOC nanosheets	Nitric oxide (NO)	Removal percentage: 50.2%	[250]
Amino-modified attapulgite (M-ATP)	Removal of Pb^2+^, and Cu^2+^ by adsorption on Amino-modified attapulgite (M-ATP)	Pb^2+^, Cu^2+^	Pb^2+^: 53.58 mg/g Cu^2+^: 28.86 mg/g	[251]
MIL-101(Fe) and MIL-101(Fe,Co)	Removal of Ciprofloxacin (CIP) by MIL-101(Fe) and MIL-101(Fe,Co)	Ciprofloxacin (CIP)	Removal percentage: 97.8%	[253]
CuCo_2_O_4_/BiVO_4_	Removal of 4-Nitrophenol	4-Nitrophenol	Not supplied	[254]
MOF-545	Removal of lead by adsorption on (MOF-545)	Pb(II)	Pb(II): 73 mg/g	[217]
Zn/Fe LDH	Removal of oxytetracycline hydrochloride (OTC) by adsorption on Zn/Fe LDH	Oxytetracycline hydrochloride (OTC)	Removal percentage: 77.23%	[255]
MoS_2_	Removal of uranyl ions U(VI) by adsorption on MoS_2_	U(VI)	MoS_2_ nanosheets: 45.7 mg/g MoS_2_ nanoflowers: 37.1 mg/g	[150]
MoS_2_ nanosheets	Removal of Pb^2+^ in aquatic systems by MoS_2_ nanosheets	Toxic metals (Pb^2+^), (Cd^2+^),	Pb^2+^: 638 mg/g under 1 sun illuminations, 902 mg/g under 4 sun illuminations Cd^2+^: 489 mg/g under 1 sun illuminations, 719 mg/g under 4 sun illuminations	[151]
PVP/MoS_2_	Removal of uranyl ions by adsorption on PVP/MoS_2_	U(VI)	U(VI): 117.9 mg/g	[215]

**Table 11 membranes-12-00360-t011:** Decomposition rate/degradation efficiency of titanium dioxide (TiO_2_).

Nanocomposite Material	Role of TiO_2_	Degraded Material	Decomposition Rate (min^−1^)/Degradation Efficiency (%)	Reference
C, N, F/TiO_2_NTs	High photocatalytic activity under UV-light	Methyl orange	Under UV-light: TiO_2_NTs: 60% C/TiO_2_NTs: 100% Under simulated sunlight: N,F/TiO_2_NTs: high activity TiO_2_NTs: low activity (high bandgap) C/TiO_2_NTs: high activity	[197]
iN-Ti_3_C_2_/TiO_2_ hybrid	Achieved a high photocatalytic performance in degrading MB.	Methylene blue (MB)	Under UV-light:0.02642 min^−1^	[198]
TiO_2_ nanoflowers(TNFs)	high photocatalytic performance for the degradation of diverse phenolic organic contaminants	Bisphenol A (BPA), diphenyl phenol, P-tert-butyl phenol, and resorcinol	Under UV-light:>95%	[199]
(TiO_2_/TiNTs)	TiO_2_/TiNTs showed about 10 times higher degradation for phenanthrene compared to the unmodified TiNTs	Cu(II), phenanthrene	Cu(II) adsorption capacity: 115.0 mg/g Under UV-light: Removal of >95% phenanthrene	[200]
Cr-TiO_2_ supported on Fe_3_O_4_	High photocatalytic activity under solar radiation	Malachite green dye (MG), total organic carbon (TOC)	Under solar radiation: 100% removal of MG 60% removal of TOC	[242]

**Table 12 membranes-12-00360-t012:** Decomposition rate/degradation efficiency of carbon nanomaterials.

Nanocomposite Material	Role of Carbon Nanomaterials	Degraded Material	Decomposition Rate (min^−1^)/Degradation Efficiency (%)	Reference
Heterogeneous Fenton catalysts (CNTs/Fh)	Degradation of bisphenol A	bisphenol A	3% CNTs/Fh: 79.1%	[116]
Co_3_O_4_/CNTs	Degradation of norfloxacin (NX)	NX	97.5%	[231]
ZIF-67 Carbocatalysts, Nitrogen-doped magnetic carbon (Co@N-C)	Degradation of BPA	BPA	60%	[235]
Ternary nanocomposites of Fe_3_O_4_ nanoparticles@ graphene–poly-N-phenylglycine nanofibers	Adsorption of Cu^2+^	Cu^2+^	95%	[241]
SWCNTs	Degradation of pharmaceutical: PhACs, ibuprofen (IBP) and sulfamethoxazole (SMX)	ibuprofen and sulfamethoxazole	At pH = 3.5: 99% for IBP and SMX	[137]
CF/BiOBr/Ag_3_PO_4_ cloth	Degradation of tetracycline hydrochloride (TCH)	TCH	90%	[154]
(N-rGO/BiVO_4_)	Degradation of methylene blue (MB)	MB	99.3%	[193]

**Table 13 membranes-12-00360-t013:** Decomposition rate/degradation efficiency of metal oxides.

Nanocomposite Material	Role of Metal Oxide	Degraded Material	Decomposition Rate (min^−1^)/Degradation Efficiency (%)	Reference
ZnO@C	Photocatalytic degradation of methylene blue	Methylene blue (MB)	99.8%	[194]
Cerium zirconium oxide (CexZryO_2_)	Photocatalytic degradation of sulfonamides	Sulfonamides	91.33%	[195]
ZnO/Al_2_O_3_	Wastewater treatment	Methyl orange dye (MO), TOC	TOC: 80.4%	[196]
NiO nanobelt	Removal of organic pollutants such as RhB, MO, MB, and CV	Removal of organic pollutants	RhB: 89% CV: 76.7% MB: 82.7% MO: 79.1%	[204]

**Table 14 membranes-12-00360-t014:** Decomposition rate/degradation efficiency of other nanocomposites.

Nanocomposite Material	Role of the Nanocomposite	Degraded Material	Decomposition Rate (min^−1^)/Degradation Efficiency (%)	Reference
Black phosphorus quantum dots/Tubular g-C_3_N_4_ (BPQDs/TCN)	Facilitates the charge spatial separation in the photocatalytic process which improves the process efficiency	Oxytetracycline hydrochloride, hexavalent chromium reduction	Oxytetracycline hydrochloride: 0.0276 min^−1^, Hexavalent chromium: 0.0404 min^−1^	[201]
CdSe-Ag-WO_3_-Ag photocatalyst	Strong redox capacity, enhanced optical absorption and accelerated transfer and separation of carriers	Cefazolin (CFZ)	CFZ: 96.32% in 30 min	[243]
Sodium titanate nanotubes (Na-TNT)	Photocatalytic degradation of nickel (Ni(II)), methylene blue (MB)	Nickel (Ni(II)), methylene blue (MB)	90% of Ni(II) ions within the first 15 min. Removed 99.5% of MB	[202]
Fe_2_O_3_-PC nanohybrids	Photocatalytic degradation of methylene blue (MB)	Methylene blue (MB)	Fe_2_O_3_: Removed 56% of MB Fe_2_O_3_-PC: Removed 75% of MB	[203]
Carbon dots/g-C_3_N_4_ (C-CN) heterostructures	Photocatalytic degradation of sulfamethoxazole (SDZ)	Sulfamethoxazole (SDZ)	0.5C-CN: 62.7% 1C-CN: 75.2% 3C-CN: 92.8% 5C-CN: 85.7%	[205]
AgBr/h-MoO_3_ composite	Photocatalytic degradation of trimethoprim (TMP)	Trimethoprim (TMP)	TMP: 97%	[206]
CN-CGs	Photocatalytic degradation of Total organic carbon (TOC), bisphenol A (BPA)	Total organic carbon (TOC), bisphenol A (BPA)	BPA: 90% TOC: 80%	[207]
Fe/Fe_3_C@PC	Photocatalytic degradation of sulfamethazine (SMT)	Sulfamethazine (SMT)	SMT: 99.2%	[236]
N-doped BiVO_4_	Photocatalytic degradation of ibuprofen (IBP)	Ibuprofen (IBF)	IBP: 90%	[208]
Bi/Fe0	Photocatalytic degradation of hexahydro-1,3,5-trinitro-1,3,5-triazine (RDX)	hexahydro-1,3,5-trinitro-1,3,5-triazine (RDX)	RDX: The best RDX degradation was achieved using 4%-Bi/Fe0 (atomic ratio) NPs	[244]
PPECu thin film electrode	Photocatalytic degradation of phenol and 2,4-DCP	Phenol, 2,4-DCP	Photocatalytic degradation of phenol and 2,4-DCP activity was 2.52 and 3.85 times higher than g-C_3_N_4_	[209]
Fe_x_Mo_1-x_S_2_ catalysts	Photocatalytic degradation of propranolol	Propranolol	90% at pH = 4.0	[210]
P-doped porous g-C_3_N_4_	Photocatalytic degradation of rhodamine B (RhB)	Rhodamine B (RhB)	RhB: 99.5%	[211]
1D/2D W_18_O_49_/g-C_3_N_4_ nanocomposites	Photocatalytic degradation of ibuprofen (IBF)	Ibuprofen (IBF)	IBF: 96.3%	[212]

## Data Availability

Not applicable.

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
