# Peer review of "A Literature Review of Modelling and Experimental Studies of Water Treatment by Adsorption Processes on Nanomaterials"

_membranes, 2022, doi:10.3390/membranes12040360_

Round 1
Reviewer 1 Report
A literature review of modelling and experimental studies of water treatment by adsorption processes on nanomaterials
Reviewer’s comments:
This work presents a critical review of the latest experimental and simulation methods on wastewater treatment by adsorption on nanomaterials for the removal of pollutants such as bacteria and heavy metals. The review includes the wastewater treatment processes that were carried out using membranes and nanoparticles. But the manuscript lacks explanation regarding major techniques/processes and characterization analysis for wastewater treatment. Not ample explanation regarding the treatment of dyes, pesticides, bacteria, heavy metals have been given. Moreover, the manuscript lacks coherence in writing of paragraphs, has repetition of lines and lacks some major figures.
Here are some major suggestions
- The introduction section should incorporate some more insights into the development of adsorption enhanced photocatalysts based systems/membranes for wastewater treatment
- The whole review is restricted to some particular nanomaterials for the synthesis of adsorbents/membranes like GO, MoS2, TiO2, and CNTs only. The authors need to explain the majority of metal oxides-based nanocomposites (binary, ternary, quaternary), polymers, and other supports and adsorbents/photocatalysts being used as membranes for wastewater treatment
- The authors need to provide a schematic diagram for the nanomaterials being used for the synthesis of membranes as adsorbents/photocatalysts to give an overall outline of review
- Authors need to explain the nature and synthesis methods of nanomembranes in text too with a schematic diagram. For a review, a mere single paragraph explanation on this wide topic is not enough. The authors need to explain this in detail
- In the topic 2.2 nanomaterials, the paragraph should explain the nature of nanomaterials like semiconductor metal-oxides-based composites, heterojunctions formation, doping, support materials. Besides, their properties explanation like bandgap that makes them suitable for adsorption and photocatalysis should also be present in detail
- Wastewater treatment is a multi-dimensional topic that encompasses the removal of dyes, pesticides, heavy metals and bacteria. Authors should explain recent previously published reports regarding adsorbent/photocatalysts membranes
- The authors need to add a figure explaining a detailed mechanism of adsorption and removal of pollutants by carbon/zeolite-based nanomembranes
- In the photocatalysis section, authors need to explain Fenton, photo Fenton, heterogeneous photocatalysis, ozonolysis processes
- Authors should also add 1-2 comprehensive pages regarding the essential characterization techniques for evaluating the synthesis of adsorbents/nanomaterials
- In Fig.3 authors need to explain the oxidation-reduction processes being carried out in visible light catalysis
- In paragraph 3.2, authors should start explaining with nature of metal oxide semi-conductor materials, their nature as binary and ternary compounds
- Tables 3,4, 5, and 6 should be more comprehensive with model pollutants, morphology, removal efficiencies, light sources, etc.
- The authors have not explained the kinetic modeling of adsorption/photocatalysis. Authors need to explain the various kinetic models of adsorption and photocatalysis
- The conclusion should be more coherent and precise
Reviewer 2 Report
The paper “A literature review of modelling and experimental studies of water treatment by adsorption processes on nanomaterials” is interesting but need a lot of work to streamline and improve the present content.
This is a paper for a renowned journal; I wonder why authors are calling it a chapter many times (as if it was prepared for a chapter earlier)
Abstract shall contain the most significant findings of this article to readers,
Why it is precisely indicating “removal of pollutants such as bacteria and heavy metals” in abstract, it should not be that specific in abstract when we are presenting for wastewater in title
Kindly explain -Line 40-41 In other cases, when the consumption of surface water sources is very high, such as rivers and lakes, the decrease in water levels leads to an increase in the concentration of minerals and pollutants [7].
Figure 1 is not required
Correct the citation of chemical formula - molybdenum disulphide (MoS2), TiO2, and various other throughout manuscript
Fig.3 quality very poor
Line 135-136, kindly correct statement “Therefore, the adsorption process is one of the most important water filtration processes”
Why the authors want to talk about adsorption from line 135 onwards, when the topic follows the next heading
More detail about the fabrication and characterization of the nanocomposite materials experimentally and by simulation are given in Chapter 2. ?????? define chapter (why this term)
Similarly line 494
Number of language errors line 253
It would be good to add a column in the last of table 1 for the type of pollutant treatment used?
Illustrations must be very selective and informative of high significance
Conclusions section is long and shall be concise information as a short summary of the manuscript for the most significant findings
Reviewer 3 Report
This review article is interesting, but the literature is not well analyzed. Advanced modeling via physical models of adsorption process should be detailed in the revised version. Please see these references (please note that is not necessary to use these references to improve this review, the important point is to add new part of physical models application).
New physicochemical interpretations for the adsorption of food dyes on chitosan films using statistical physics treatment. Food Chemistry.
Novel insights into the adsorption mechanism of methylene blue onto organo-bentonite: Adsorption isotherms modeling and molecular simulation, Journal of Molecular Liquids
Round 2
Reviewer 1 Report
The author has incorporated most of the changes and the document can be accepted after improvement in English language.
Reviewer 3 Report
Accept